# Identification of Hyperelastic Material Parameters of Elastomers by Reverse Engineering Approach

**DOI:** 10.3390/ma15248810

**Published:** 2022-12-09

**Authors:** Burak Yenigun, Elli Gkouti, Gabriele Barbaraci, Aleksander Czekanski

**Affiliations:** Department of Mechanical Engineering, York University, Toronto, ON M3J 1P3, Canada

**Keywords:** reverse engineering, artificial neural network, hyperelasticity, rubber

## Abstract

Simulating the mechanical behavior of rubbers is widely performed with hyperelastic material models by determining their parameters. Traditionally, several loading modes, namely uniaxial tensile, planar equibiaxial, and volumetric, are considered to identify hyperelastic material models. This procedure is mainly used to determine hyperelastic material parameters accurately. On the contrary, using reverse engineering approaches, iterative finite element analyses, artificial neural networks, and virtual field methods to identify hyperelastic material parameters can provide accurate results that require no coupon material testing. In the current study, hyperelastic material parameters of selected rubbers (neoprene, silicone, and natural rubbers) were determined using an artificial neural network (ANN) model. Finite element analyses of O-ring tension and O-ring compression were simulated to create a data set to train the ANN model. Then, the ANN model was employed to identify the hyperelastic material parameters of the selected rubbers. Our study demonstrated that hyperelastic material parameters of any rubbers could be obtained directly from component experimental data without performing coupon tests.

## 1. Introduction

Elastomers, also known as rubbers, are widely preferred due to their complex mechanical behavior and long-term resistance to extreme conditions such as large deformation, various strain rates, and high temperatures [1,2]. They are mainly used in additive manufacturing, electronics, construction, and biomedical fields. Selecting a suitable elastomer is a complex procedure that requires the characterization of mechanical response to deformation under specific environmental conditions. Hence, it is necessary to incorporate conditions that simulate the application-specific behavior.

Rubbers are considered materials that exhibit nonlinear hyperelastic and viscoelastic behavior [3,4,5,6,7,8]. Several experimental tests are required to characterize their mechanical behavior, which can be categorized into two groups: applying loading/unloading and holding constant deformation for a specific period (input). Elastomers’ response to those conditions is increasing/decreasing deformation and stress relaxation, respectively (output). Several hyperelastic models (e.g., Ogden, Neo-Hookean, Mooney–Rivlin, Arruda Boyce) have been developed for the first group, where a specimen is subjected to loading or unloading, and their parameters are required to be determined [9,10,11,12]. Appropriate model selection based on strain energy density functions depends on the selected rubber, the deformation range, and the environmental conditions (e.g., temperature) [13,14,15]. Their stress–strain curves provided by experimental data ensure the best fitting. For the second group, stress relaxation can be modeled using the Prony series for multiple terms, where parameters must be determined [16,17].

Other factors determining the hyperelastic material model include the selected model’s stability for an application representative deformation range and the friction force. Evaluating and choosing stable model(s) can be determined by Drucker’s stability criteria [18,19]. Considering all those factors, selecting the appropriate application-specific rubber to simulate multiple conditions accurately is possible. However, the experimental setup’s complexity and additional labor requirements to perform tests are some limitations of the hyperelastic material modeling and avoid including friction coefficients. Moreover, obtaining a perfect model to fit experimental data is challenging since the analytical procedure of determining material parameters involves solving a set of equations.

Thus, the traditional procedure [10,20,21,22] mentioned above is time- and money-consuming as it requires many experiments that might need to be repeated if the selected rubber is not adequate for application. Third-party software or modules based on a curve-fitting principle are often used to identify hyperelastic material parameters. On the other hand, in recent decades, a reverse engineering approach using commercial software for various applications has been widely embraced [23,24,25,26,27,28,29,30,31,32,33]. The iterative finite element model (FEM), machine learning, and virtual fields method (VFM) are the most known reverse engineering approaches [34]. The iterative FEM approach is based on finding the minimum error between experimental and predicted data by performing iterative finite element analysis (FEA) using estimated initial material parameters. FEA is simulated by changing the initial material parameters in each iteration according to a chosen optimization technique, and iteration continues until it reaches minimum error. For example, there is a study using this approach in which visco-hyperelastic parameters of brain white matter were predicted [25]. Iterative compression FEA was simulated to obtain the optimal constitutive model parameters, and a genetic algorithm was used to match the error between the experimental and FEA results. Another study used an optimization loop designed in commercial software (Isight) to identify the viscoelastic material parameters of filled rubber compounds [27]. Material parameters were determined by calibrating the data from uniaxial stress–strain tensile, volumetric, and stress relaxation tests. In another example, nonlinear visco-hyperelastic material parameters of homo-polymer were determined using the parallel rheological framework (PRF) model in software [30]. However, a custom optimization script or third-party software is required for optimization in this approach. Moreover, choosing the wrong optimization technique may cause finding a local minimum point instead of a global minimum point, or optimization may take longer since FEA is simulated in each iteration [35].

To overcome this considerable drawback, an artificial neural network (ANN) can be employed to characterize elastomers, which involves an educated guess-and-check process. ANN is a powerful tool for analyzing data and creating complex relationships between variables [36,37,38]. Therefore, some researchers used an ANN model to accelerate or predict material properties [33,39,40,41,42,43,44,45,46]. This indirect approach creates an ANN model from the FEA dataset to accelerate optimization instead of using FEA in each iteration. FEA is performed by generating a dataset from random material parameters to build the ANN model. There are several attempts in relation to the ANN approach. For example, the hyperelastic parameter of silicone rubber was identified using the approach mentioned above [26]. Uniaxial tensile FEA using predefined parameters was simulated to create a database for the general regression neural network (GRNN) model. Once the GRNN model was created, the hyperelastic material parameters of silicone rubber were predicted using a GRNN model and optimization technique. Another study estimated different hyperelastic material parameters of Neodymium Butadiene rubber (NdBR), Ogden-3, Neo-Hooken, and Mooney–Rivlin using neural networks [47]. In their study, strain invariants were used to train an ANN model, and hyperelastic material parameters validation was performed through uniaxial and equibiaxial testing. It is worth mentioning that there are some challenges in employing the ANN technique. For example, if data sets are created with a complete factorial method in this approach, it causes problems such as overfitting in the ANN model and poor nonlinear relationships between data

The virtual fields method (VFM) consists of an analytical solution of a constitutive model, digital image correlation, and optimization method. Virtual fields, which can be strain or displacement, are predicted by solving a constitutive analytical model with guess parameters. Then, the virtual prediction is compared to digital image correlation measurements through a cost function to be minimized [48,49]. This approach is applied to determine the hyperelastic material parameters of rubber specimens with a cruciform shape [34,50,51,52]. Similar to the previously described approaches, the selection of the initial parameters and optimization techniques directly affects the quality of the prediction in this approach. Moreover, these indirect approaches are material-dependent, and the procedure must be repeated for another material. The current study identifies different rubbers’ responses for application-specific component geometry that provides O-ring shapes applying a direct material-independent reverse engineering approach. Our goal is to determine the hyperelastic material parameters that will fulfill the application requirements and lead to a suitable selection of rubber. Specifically, we aim to identify the hyperelastic material parameters of different rubbers (silicon, neoprene, and natural rubber) from the developed ANN model without performing experimental coupon tests.

Therefore, this study aims to eliminate the requirement of performing numerous experimental tests, optimizing errors and solving analytical equations for each selected rubber by (a) developing an ANN model, (b) identifying the hyperelastic material parameters as well as friction coefficients, and (c) validating the predictions. Compared to the previous studies in developing relevant models for material parameter identification, we develop an ANN model to determine the hyperelastic material parameters of any rubber for any temperature or surface conditions by eliminating iterative FEA simulations or optimization methods. The ANN model is based on application geometry that specifically gains accurate results. Nevertheless, our study can be employed in other geometry-specific cases and any hyperelastic material model (other than Ogden-2). The flow chart of this study is shown in Figure 1.

## 2. Experimental Study

### 2.1. Samples Configurations

#### 2.1.1. Component Test Configurations

Neoprene rubber, natural rubber, and silicone rubber batches with thicknesses of 25.4 mm, 3.175 mm, and 1.5875 mm supplied from McMaster-Carr were used in this study. Rubbers batches with a thickness of 25.4 mm, were cut with a water jet to outer/inner diameters of Ø127/Ø101.6 mm, Ø101.6/Ø50.8 mm, and Ø50.8/Ø25.4 mm for O-ring tension, O-ring compression, and O-ring multi-contact tests, respectively. Neoprene rubber samples for O-ring tests are shown in Figure 2 and Figure 3.

#### 2.1.2. Coupon Test Configurations for Validation Study

Rubber batches with a thickness of 3.175 mm for uniaxial and equibiaxial tests and rubber batches with a thickness of 1.5875 mm for planar tests were cut with a die. Silicone rubber coupon test samples are shown in Figure 4. Coupon test sample dimensions can be found in our previous study [53].

### 2.2. Experimental Setup

#### 2.2.1. Experimental Component Testing for Material Predictions and Validation Study

The selected components were cut in O-ring shapes, as shown in Figure 2, and subjected to tension and compression, as shown in Figure 5. We performed quasi-static tension and compression with the crosshead moving at 1 mm/s speed with a sampling rate of 1 Hz using a universal MTS Criterion 43 machine. The component tests were performed three times for consistency. Experimental component results of O-ring tension and O-ring compression were used to predict hyperelastic material parameters from the developed ANN model. Additionally, the O-ring compression test contributed to estimating the friction coefficient more accurately. The O-ring multi-contact component test results, shown in Figure 6, were used to validate the ANN-predicted hyperelastic material parameters.

To demonstrate the accuracy of the developed ANN approach, we simulated different conditions. Thus, we aimed to predict the behavior of neoprene’s component when subjected to O-ring multi-contact tests at various environmental conditions (not only the ambient). To compare the outcomes of the ANN approach, actual test data were required; hence, we performed O-ring tension and O-ring compression of the neoprene test (Figure 2) with the setup shown in Figure 5. We repeated the same experimental procedure at the elevated temperatures of 50 °C and 80 °C and compared them with the corresponding FEA results.

#### 2.2.2. Experimental Coupon Testing for The Validation Study

Rubbers exhibit a hyperelastic response to an applied deformation; their behavior can be modeled using their energy density function. Many numerical models have been defined previously, requiring the determination of their material parameters, which is feasible by performing experimental tests. Rubbers’ response must be tested under various deformation directions; hence, uniaxial, planar, and equibiaxial tests are usually required. The curve fitting procedure is followed to determine the materials parameters of the selected hyperelastic model. The material parameters defined by the procedure mentioned above can be used as an input for FEA, which is considered a “traditional approach” to simulating rubbers’ behavior.

In our study, uniaxial, planar, and equibiaxial coupon tests were carried out at a quasi-static loading rate with a crosshead moving at 1 mm/s with a sampling rate of 1 Hz at room temperature using a universal MTS Criterion 43 machine. An MTS LX500 laser extensometer was used to measure strain during the tests. A 500 N load cell was used for all tests except equibiaxial coupon tests. Since equibiaxial coupon tests require more than a 500 N load cell, a 10,000 N load cell was used for equibiaxial coupon tests. All experiments were repeated three times to ensure consistency. The experimental coupon testing setups are shown in Figure 7.

## 3. Numerical Modelling

### 3.1. Traditional Approach to Determining Hyperelastic Material Parameter

The commonly used procedure of simulating the mechanical behavior of rubbers in different conditions is running an FEA based on coupon experimental test data. Namely, the hyperelastic model was built based on experimental data (uniaxial, planar, and equibiaxial). In our study, we used this approach (from now on, mentioned as a “traditional approach”) to predict rubber-specific and application-specific applications. Specifically, we determined the material parameters of the Ogden-2 hyperelastic model by performing coupon testing of uniaxial, planar, and equibiaxial experiments as described previously (Figure 7). Following the curve fitting procedure for the obtained test data results, we determined the material parameters (mu1, alpha1, mu2, and alpha2) using Abaqus as input for our FEA simulations.

### 3.2. ANN Approach to Determine Hyperelastic Material Parameter

In addition to the previous ways to gain the mechanical response of rubbers to deformation, we decided to include A.I. benefits in our study by creating an ANN path. In addition to the previous (traditional) approach, the ANN approach requires no experimental coupon tests to determine the hyperelastic material parameters. Since we created virtual data sets of 100 different cases of coefficients, we used them as input to train an ANN model, which was then used for FEA validation simulations. It must be mentioned that the build ANN model corresponds to geometry-specific applications, which for our study refers to O-ring tension and O-ring compression deformation of rubbers.

#### 3.2.1. Data Derivation

Apart from the traditional approach of determining hyperelastic material parameters, we decided to follow another path by creating virtual data sets. Initially, we selected the most accurate model for O-ring shapes, the Ogden hyperelastic model, with two terms (noted as Ogden-2) including four parameters: alpha1, mu1, alpha2, and mu2 [54,55,56]. Regarding the Ogden material model, the strain energy density is expressed in Equation (1) in terms of the principal stretches $\lambda_{1},\lambda_{2},\;and\;\lambda_{3}$ [57].
(1)$$W=\overset{N}{\underset{i=1}{\sum }}\frac{2*\mu_{i}}{\alpha_{i}^{2}}*({\bar{\lambda }}_{1}^{\alpha_{i}}+{\bar{\lambda }}_{2}^{\alpha_{i}}+{\bar{\lambda }}_{3}^{\alpha_{i}}-3),$$
where, $\mu_{i}$ and $a_{i}$ are material parameters.

Based on this model, we run FEA using commercial software (ABAQUS/Standard-2019) for the geometry-specific cases of O-ring shapes. Specifically, O-ring tension and O-ring compression FEA models were simulated by applying 75 mm and 15 mm displacements, respectively. O-ring tension and O-ring compression FEA models are shown in Figure 8. An 8-node linear brick, hybrid, constant pressure, reduced integration, hourglass control, C3D8RH, was selected as the element type. Mesh sizes for O-ring tension and O-ring compression were selected as 1.25 and 0.625, respectively.

O-ring FEA simulations were repeated 100 times using 100 randomly generated data sets. With this method, we also included the virtual friction parameter of rubbers, which is only possible experimentally by performing additional testing. The randomly created hyperelastic material parameters and friction coefficients for 100 cases are shown in Table 1.

In order to guarantee that the resulting material parameters provide stable models, we created Abaqus scripts for evaluating the randomly created parameters (Table 1). The outcomes showed stability for all cases according to Drucker’s stability criteria. The corresponding virtual curves of force and displacement obtained from the O-ring tension and O-ring compression FEA simulations are shown in Figure 9. It must be noted that these virtual data sets correspond to the geometry-specific application (O-ring tension and O-ring compression) but are material-independent; namely, they can be used for any rubber, as will be proved in the following sections.

#### 3.2.2. ANN Model Development

The force-displacement curves of the virtual data sets (100 cases) corresponding to the geometry-specific simulation, shown in Figure 9, were used to build the ANN in the commercial software of MATLAB 2021a. Specifically, the force values obtained from the O-ring tension and O-ring compression FEA simulations were selected as inputs, and Ogden-2 hyperelastic material parameters were selected as outputs. The same data quantities were taken from the O-ring tension and O-ring compression FEA simulations to prevent overfitting and obtain uniform distribution in the ANN model. The force values at each 15 mm displacement in the virtual O-ring tension FEA results and the force values at each 3 mm displacement in the virtual O-ring compression FEA results were used for the ANN training database. The ANN structure is shown in Figure 10.

Since it is not intended to determine the friction coefficient of rubbers in this study experimentally, it was added as an output parameter to the ANN model, assuming it is equal for both O-ring tension and O-ring compression tests. The developed ANN model contains fully connected layers: one input layer, two hidden layers, and one output layer. Levenberg–Marquardt’s backpropagation algorithm was selected as the training function, and the hyperbolic tangent sigmoid transfer function, tansig, was selected as the activation function. The properties of the ANN model are provided in Table 2.

## 4. Results

### 4.1. ANN Material Predictions

Ogden-2 hyperelastic material parameters of natural, silicone, and neoprene rubbers were predicted by the ANN model and determined by the “traditional approach” in Table 3. The friction coefficients of the rubbers were predicted in line with the literature by the ANN model and used for calculating friction force [58,59,60].

For the case of neoprene rubber’s deformation at 50 °C and 80 °C, we only included ANN-predicted hyperelastic material parameters, shown in Table 4. These coefficients were used to validate neoprene rubber O-ring test results at 50 °C and 80 °C.

### 4.2. Testing of Ogden-2 Hyperelastic Material Parameters

The FEA results of O-ring tension and O-ring compression tests, using the ANN-predicted and ‘traditional approach’ determined by hyperelastic material parameters, are plotted with experimental results in Figure 11 for rubbers (neoprene rubber, natural rubber, and silicone rubber). It is evident that the FEA results, simulated by Ogden-2 hyperelastic material parameters from both methods, exhibit no significant difference for O-ring tension and O-ring compression experimental results. This observation is common for all rubbers. Additionally, FEA results, simulated by ANN-predicted hyperelastic material parameters, demonstrated high accuracy with experimental results for all rubbers.

### 4.3. Validation of Ogden-2 Hyperelastic Material Parameters

#### 4.3.1. Validation of Component Test: O-Ring Multi-Contact

ANN-predicted Ogden-2 hyperelastic material parameters of neoprene rubber, natural rubber, and silicone rubber were validated using the O-ring multi-contact component test. Validation of the O-ring multi-contact results for the selected rubbers is shown in Figure 12. The developed-ANN model provided accurate estimates for all selected rubbers.

#### 4.3.2. Validation of Coupon Test

FEA coupon (uniaxial, planar, and equibiaxial) test simulations were also used to validate selected rubbers’ ANN-predicted Ogden-2 hyperelastic material parameters. The FEA results, using hyperelastic material parameters predicted by the ANN model and determined with the traditional approach, were compared with the experimental coupon test results of these rubbers in Figure 13. Figure 13 presents the accuracy of the Ogden-2 model for selected rubbers, and it can be claimed that Ogden hyperelastic material model was successfully applied to the analysis of O-rings.

### 4.4. Case Study

To evaluate the ability of the developed ANN model to predict the hyperelastic material parameters at different temperatures, we selected neoprene rubber components to be tested and simulated when subjected to O-ring multi-contact deformation at 50 °C and 80 °C. Figure 14 shows the results of neoprene subjected to O-ring tension, O-ring compression, and O-ring multi-contact experimental tests compared to FEA predictions at those temperatures. It must be noted that O-ring tension and O-ring compression FEA database results are independent of temperature variations (e.g., 50 °C and 80 °C). ANN-predicted hyperelastic material parameters provide high accuracy, similar to the corresponding O-ring tension and O-ring compression experimental results.

## 5. Conclusions

We developed an ANN model to identify hyperelastic material parameters in the current study. The developed ANN model was applied for geometry-specific case studies (at the component level); thus, it is material-independent as it allows the prediction of hyperelastic materials parameters for any rubber. The ANN model was initially built using O-ring tension and O-ring compression FEA data sets. Our approach was validated on two levels: by comparing ANN predictions with (i) component-level: O-ring multi-contact component tests and (ii) coupon-level: uniaxial, equibiaxial, and planar coupon tests.

It is worth noting that the ANN model is able to predict rubbers’ hyperelastic behavior at any temperature conditions. To validate the feasibility of this characteristic, we compared the developed ANN model with the experimental results of neoprene’s O-ring components subjected to different temperature levels. The results showed that the predictions have adequate accuracy.

In conclusion, the advantage of using the ANN model to predict rubbers’ hyperelastic behavior is that it can be obtained directly from experimental component tests without performing material coupon tests. Moreover, the developed ANN is a material and temperature-independent model, as it provided accurate predictions of the hyperelastic behavior of any selected rubber exposed at different elevated temperatures. At the same time, it allows determining rubbers’ surface properties (friction) without performing extra experimental tests, which are both time- and money-consuming and challenging to achieve.

## Figures and Tables

**Figure 1 materials-15-08810-f001:**
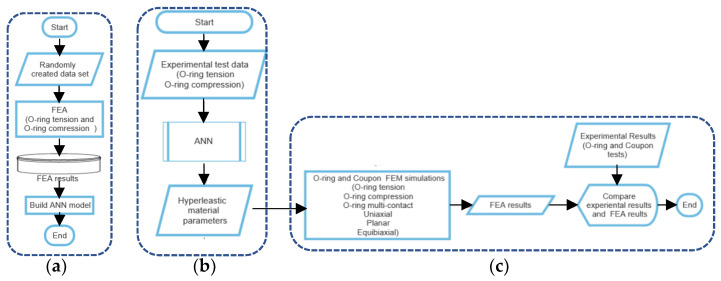
Flow charts: (**a**) ANN-subroutine, (**b**) parameter identification, and (**c**) validation steps.

**Figure 2 materials-15-08810-f002:**
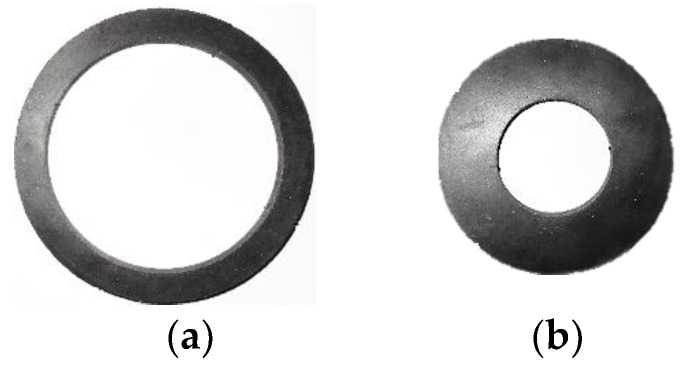
Component test samples used to determine material parameters: (**a**) O-ring tension test and (**b**) O-ring compression test.

**Figure 3 materials-15-08810-f003:**
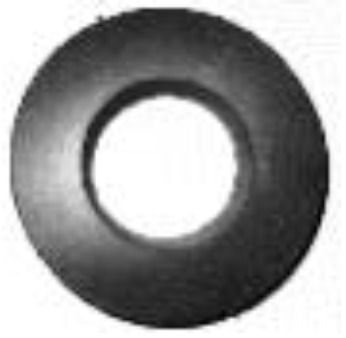
Component tests sample used to validate material parameters: O-ring multi-contact test.

**Figure 4 materials-15-08810-f004:**
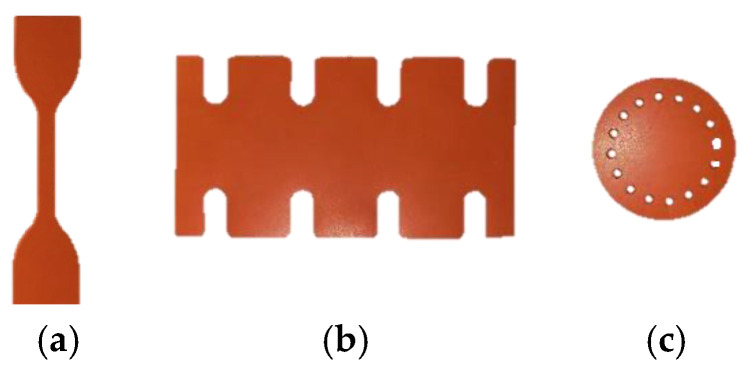
Silicone rubber coupon tests samples for validation study: (**a**) uniaxial test, (**b**) planar test, and (**c**) equibiaxial test.

**Figure 5 materials-15-08810-f005:**
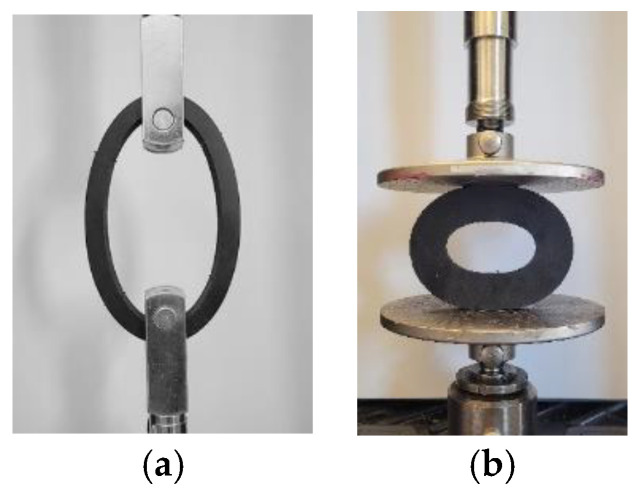
Experimental setups for material parameter determination: (**a**) O-ring tension and (**b**) O-ring compression test.

**Figure 6 materials-15-08810-f006:**
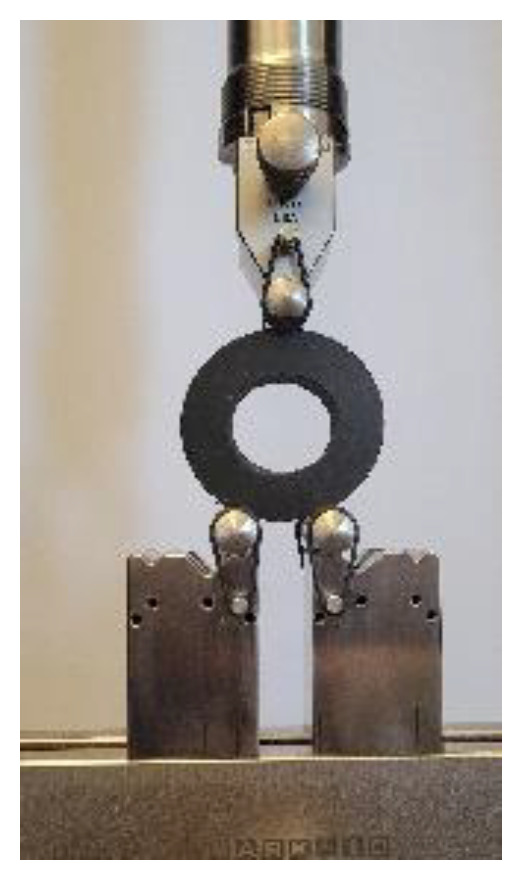
Experimental setups of O-ring multi-contact test for validation.

**Figure 7 materials-15-08810-f007:**
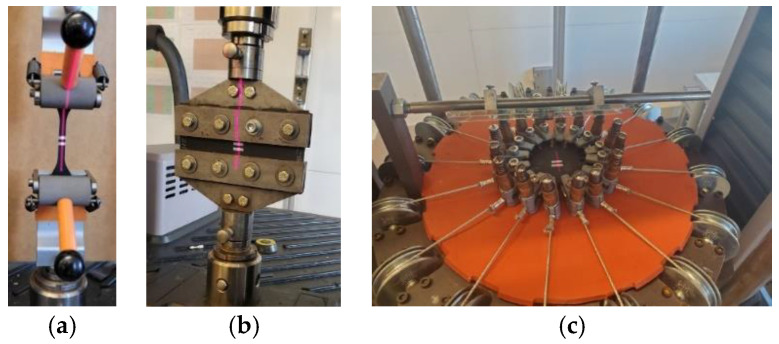
Experimental coupon testing setup: (**a**) uniaxial test, (**b**) planar test, and (**c**) equibiaxial test.

**Figure 8 materials-15-08810-f008:**
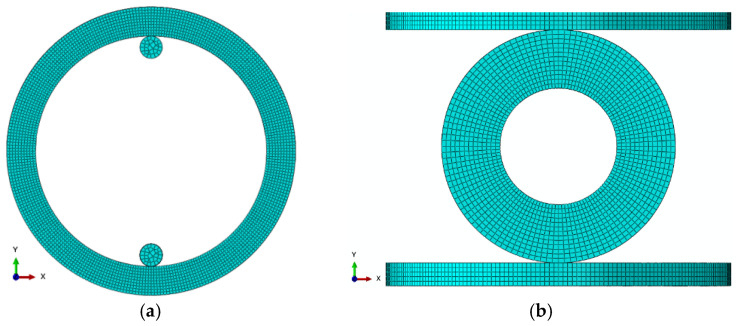
FEA model: (**a**) O-ring tension and (**b**) O-ring compression test.

**Figure 9 materials-15-08810-f009:**
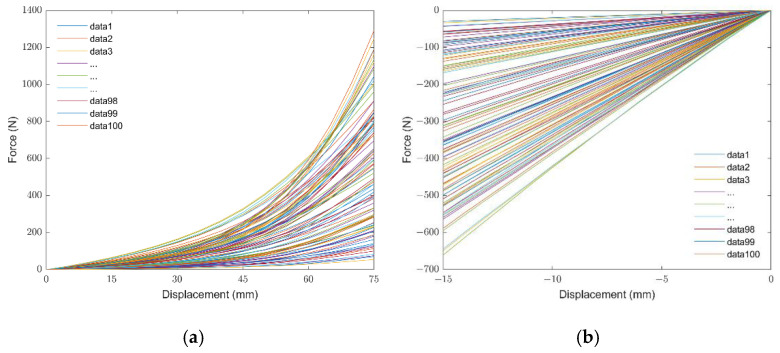
Virtual data sets: (**a**) Oring tension test and (**b**) Oring compression test.

**Figure 10 materials-15-08810-f010:**
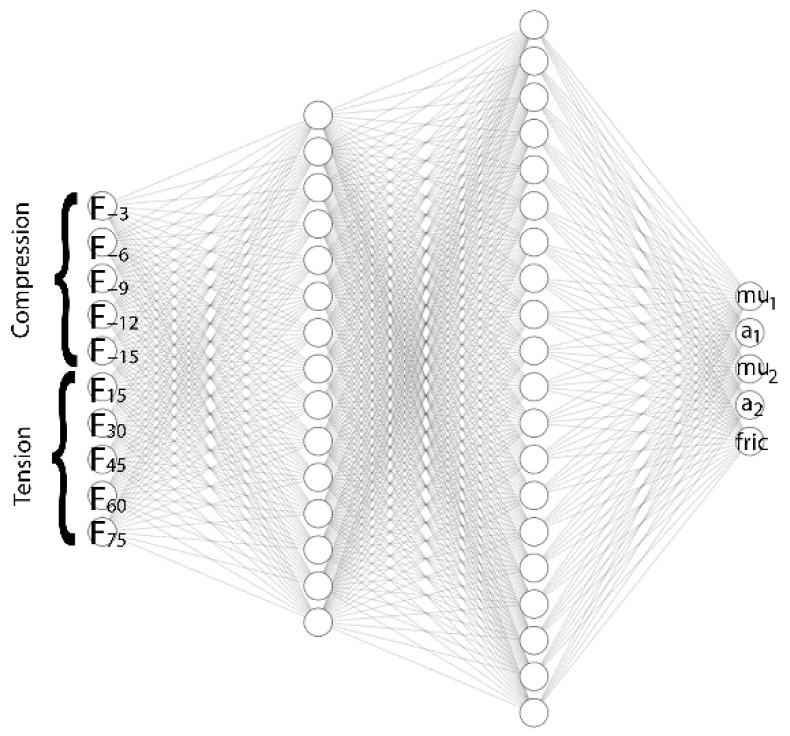
ANN structure with two hidden layers.

**Figure 11 materials-15-08810-f011:**
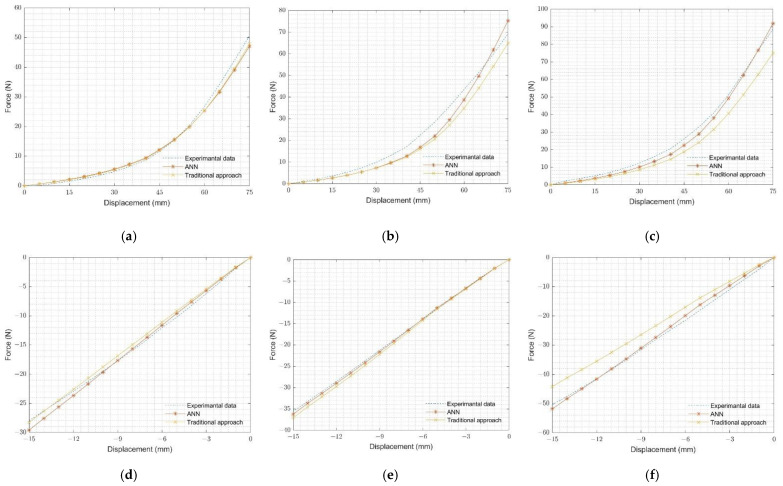
Comparison of experimental and FEA results: (**a**) neoprene rubber O-ring tension test, (**b**) natural rubber O-ring tension test, (**c**) silicone rubber O-ring tension test, (**d**) neoprene rubber O-ring compression test, (**e**) natural rubber O-ring compression test, and (**f**) silicone rubber O-ring compression test.

**Figure 12 materials-15-08810-f012:**
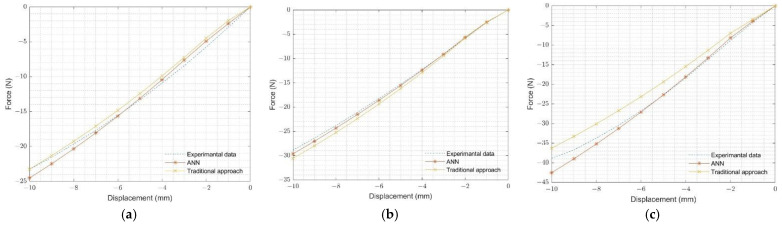
Comparison of O-ring multi-contact experimental and FEA results: (**a**) neoprene rubber, (**b**) natural rubber, and (**c**) silicone rubber.

**Figure 13 materials-15-08810-f013:**
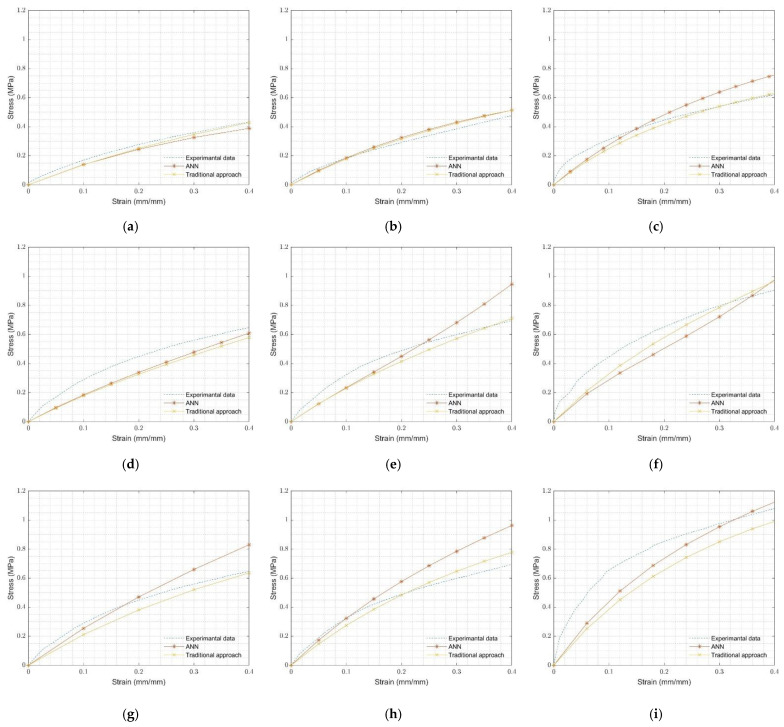
Comparison of experimental and FEA results of coupon tests: (**a**) neoprene rubber uniaxial test, (**b**) neoprene rubber planar test, (**c**) neoprene rubber equibiaxial test, (**d**) natural rubber uniaxial test, (**e**) natural rubber planar test, (**f**) natural rubber equibiaxial test, (**g**) silicone rubber uniaxial test, (**h**) silicone rubber planar test, and (**i**) silicone rubber equibiaxial test.

**Figure 14 materials-15-08810-f014:**
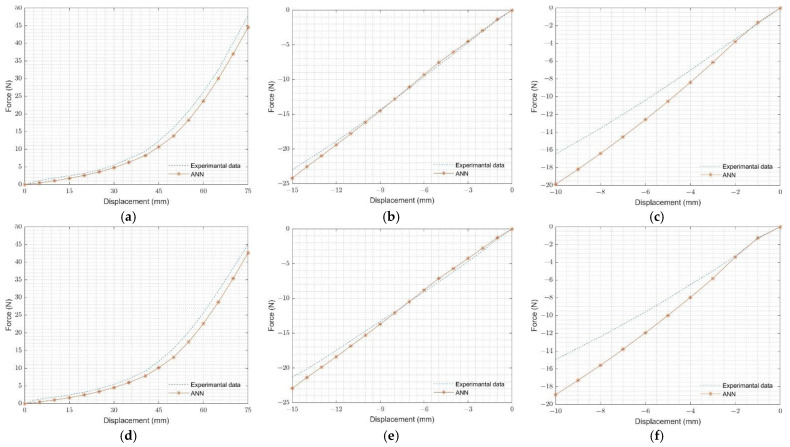
Comparison of experimental and FEA results for neoprene rubber for (**a**) O-ring tension test at 50 °C, (**b**) O-ring compression test at 50 °C, (**c**) O-ring multi-contact test at 50 °C, (**d**) O-ring tension test at 80 °C, (**e**) O-ring compression test at 80 °C, and (**f**) O-ring multi-contact test 80 °C.

**Table 1 materials-15-08810-t001:** Randomly generated material parameters.

Data Number	Alpha1	Mu1	Alpha2	Mu2	Friction
1	1.5540	2.1625	2.9479	−4.1510	1.8134
2	1.5422	4.3829	5.8699	−5.9271	0.0443
3	1.4168	3.5313	4.7432	−6.0336	1.1834
.	.	.	.	.	.
.	.	.	.	.	.
.	.	.	.	.	.
50	0.814	5.747	7.757	−6.404	0.889
51	0.553	9.692	4.545	4.358	0.950
.	.	.	.	.	.
.	.	.	.	.	.
.	.	.	.	.	.
99	0.9865	8.1362	7.2496	1.5037	1.0994
100	0.2656	5.5092	4.5852	2.8785	1.0216

**Table 2 materials-15-08810-t002:** ANN model properties.

Parameter	Value
Input layer size	10
Hidden layer1 size	15
Hidden layer2 size	20
Output layer size	5
Activation function	tansig
Backpropagation algorithm	Levenberg–Marquardt
Data ratio	%70 training, %15 validation, %15 testing

**Table 3 materials-15-08810-t003:** Ogden-2 hyperelastic material parameters based on coupon tests (traditional) and component tests (ANN).

Parameters	Natural Rubber		Silicone Rubber		Neoprene Rubber	
	Traditional	ANN	Traditional	ANN	Traditional	ANN
alpha1	0.2526	0.2422	0.6793	0.3701	0.0292	0.2427
mu1	3.8250	4.9068	1.1441	2.0343	1.4824	3.6534
alpha2	0.4098	0.4102	0.1117	0.5590	0.4833	0.2913
mu2	−0.3865	4.8452	1.1440	2.0467	1.4970	0.6877
friction	-	0.7465	-	0.8534	-	0.7538

**Table 4 materials-15-08810-t004:** ANN predicted Ogden-2 hyperelastic material parameters at 50 °C and 80 °C.

Parameters	ANN 50 °C	ANN 80 °C
alpha1	0.2913	0.2940
mu1	3.5942	3.7996
alpha2	0.1430	0.1159
mu2	−0.0340	−0.4132
friction	0.7513	0.8013

## Data Availability

Not applicable.

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
