# Peer review of "Identification of Hyperelastic Material Parameters of Elastomers by Reverse Engineering Approach"

_materials, 2022, doi:10.3390/ma15248810_

Round 1

Reviewer 1 Report

The manuscript is well designed and the analysis and reasonings are appropriate, however a revision is needed in the introduction section, it is recommended to investigate some similar research works in details, also the final paragraph of the introduction should explain the shortages of the previous literature and highlight the innovation of your work.

Reviewer 2 Report

This manuscript reported a ANN model of predicting hyperelestical materials parameters at different temperature. In comparison with FEA prediction, ANN model provided high accuracy, similar to the corresponding experimental results. It is noted that the ANN training data were much more close to the real conditions, and the FEA database was simplified to the different conditions, so the ANN predictions were more convinced and useful. The manuscript was well organized, and the results will benefit the computational materials. I suggest it can be accepted and published at present status. 

Author Response

Thank you very much for your comments.

We submitted the revised version without tracking changes, otherwise, it would be very hard to read as we made major changes in the manuscript. 

Reviewer 3 Report

Instead of current cumbersome testing process of rubber properties, this work presents an artificial neural network model (ANN) to determine the hyperelastic parameters of elastomeric materials. It seems that the starting point of this article is to put the experimental data into an established neural network model, and then run the MATLAB neural network to predict the hyperelastic properties of elastomer. However, many things are uncertain in this work. For example, what formula is used in ABAQUS simulation? What is the detail of artificial neural network? Is this model reliable only based on the data of three-type rubber? Therefore, I do not think this work is suitable to be published in Materials before a major modification.

Round 2

Reviewer 3 Report

It seems that all questions have been addressed, so I think this revised manuscript can be accepted.